# Colectomy among Fee-for-Service Medicare Enrollees Coded as DRG 330: A Potential Platform to Allow Consumer Cost Transparency?

**DOI:** 10.3390/healthcare8040529

**Published:** 2020-12-02

**Authors:** Byron D. Hughes, Christian Sommerhalder, E Martin Sieloff, Kari E. Williams, Douglas S. Tyler, Anthony J. Senagore

**Affiliations:** 1Department of Surgery, University of Texas Medical Branch, Galveston, TX 77555, USA; chsommer@utmb.edu (C.S.); dstyler@utmb.edu (D.S.T.); 2Homer Stryker MD School of Medicine, Western Michigan University, Kalamazoo, MI 49007, USA; martin.sieloff@med.wmich.edu (E.M.S.); asenagore@sbcglobal.net (A.J.S.); 3School of Medicine, University of Texas Medical Branch, Galveston, TX 77555, USA; Karwilli@utmb.edu

**Keywords:** colorectal, outcomes, DRG, Medicare

## Abstract

The use of Centers for Medicare and Medicaid Services Diagnosis Related Group (CMS-DRG) codes define hospital reimbursement for Medicare beneficiaries. Our objective was to assess all patients with comorbidities on admission who were discharged in the DRG 330 category to determine the impact of postoperative complications on Medicare costs. The 5% Medicare Database was used to evaluate patients who underwent a colectomy and were coded as CMS-DRG 330. Patients were divided into two groups: No surgical complications (NSC) and surgical complications (SC). Length of stay (LOS), complications, hospital charges, CMS reimbursement, discharge destination, and inpatient mortality were assessed. Statistical significance was set at *p* < 0.05. In total, 13,072 patients were identified. The SC group had higher inpatient mortality, a longer LOS (*p* < 0.0001) and was more likely to be discharged with post-acute care support (*p* = 0.0005). The use of CMS-DRG coding has the potential to provide Medicare fiscal intermediaries, beneficiaries, and providers with a more accurate understanding of the relative impact of their baseline health. The data further suggest that providers may benefit by more fully understanding the cost of preventive measures as a means of reducing total cost of care for this population.

## 1. Introduction

Colectomy is a commonly performed procedure in the field of general surgery and is associated with a relatively high morbidity and mortality, which supports this procedure as an important platform for assessing variation in costs per Medicare beneficiary [1]. The Medicare Severity Diagnosis Related Group (MS-DRG) coding platform has clear definitions for category assignment; therefore, it appears this could easily be adapted as a tool for payors, beneficiaries, and providers to transparently assess differences in potential costs of care between institutions. The principal focus of cost control has been to assess surrogate quality process measures that may or may not contribute to outcomes and therefore cost savings [2].

Hospital reimbursement for this procedure through Medicare is based on the MS-DRG devised by the Center for Medicare and Medicaid Services (CMS). This three-tier system adopted in 2007 reimburses hospitals by incremental costs in accordance with their DRG class: DRG 331 = $9913, DRG 330 = $15,150, and DRG 329 = $29,586. DRG 331 indicates patients without complication or comorbidity (CC), DRG 330 includes patients with at least one CC but no major complication or comorbidity (MCC), and 329 for patients with MCCs [3]. Individual patients are designated a class on hospital admission, which may be amended prior to discharge based on the post-admission course. The broad assumption of this system is that resource consumption is similar for all patients in a given category. However, for DRG 330 this assumption risks conflating the resource consumption related to specific present on admission comorbidities with those required to manage common surgical complications. This distinction regarding the relationship between various comorbidities and complications in this population has not been well studied.

Our group has previously explored this concept by defining the concept of “DRG migration” which is defined by patient assignment to a higher cost DRG due only to post admission comorbidity or complications (CC) [3]. This analysis confirmed the significant cost impact of converting a patient from DRG 331 to the more highly reimbursed DRG 330 due to surgical complications from the perspective of the payor, however, the provider was better reimbursed to manage those surgical complications. This results in a potential value gap where providers with lower complications are reimbursed less whereas providers with higher complication rates may benefit economically.

The current study expands on this concept by attempting to define payment variations among patients with significant comorbidities present on admission to warrant DRG 330 assignment and define the relative impact of surgical complications on the cost of care. Therefore, we assessed clinical outcomes and cost of care for this defined population based on: (1) the absence of surgical complications (NSC); or (2) the development of surgical complications (SC).

## 2. Materials and Methods

### 2.1. Study Design and Data

A descriptive comparison between two groups, designed as a retrospective cohort study using the 5% random sample Medicare Database from 2011 to 2014 was completed. The Medicare Provider Analysis and Review (MedPAR) file and Medicare beneficiary summary file were used. The summary file contains each Medicare beneficiary’s enrollment status, demographics, and mortality data. MedPAR data was used to identify inpatient variables among Medicare enrollees, including diagnoses, hospitalizations, inpatient procedures, charges, and payments from CMS.

### 2.2. Study Cohort

Patients included in the study were aged 65 and older who had undergone a colon resection Table 1. Elective colon resection was identified by Medicare inpatient reimbursement MS-DRG 330 (major abdominal small and large bowel surgery with comorbidity/complication). For this study, we included all patients assigned to DRG 330 due to their baseline comorbidities. Two groups of patients were then assessed based on: (1) the absence of surgical complications (NSC); or (2) the development of surgical complications (SC). All patients classified as DRG 331 or DRG 329 upon discharge were excluded.

The two groups were compared for LOS, hospital charges, and CMS payments. Charges in this study referred to the costs of all services provided; whereas Medicare payments referred to the amount covered by CMS. We examined comorbidities and complications using the Elixhauser Comorbidity Index. Additionally, common operative complications after elective colon resection such as ileus, surgical site infections, hemorrhage, and abscess/anastomosis leak were assessed for the SC group.

Demographic characteristics included were age, race/ethnicity, and gender. Further, race/ethnicity was classified into non-Hispanic white, non-Hispanic black, Hispanic, and Other. Elixhauser Comorbidity Index, discharge destination (skilled nursing facility [SNF], home health services [HHS], and other), and inpatient mortality were analyzed as well.

### 2.3. Statistical Analysis

Descriptive statistics were utilized to analyze patient characteristics, procedure type (open resection vs. laparoscopic resection), as well as a comparison of post-admission complication frequency between DRG 330 groups. Chi-square and Fisher’s exact tests were used for categorical variables and Welch’s *t*-test for continuous variables. An unadjusted analysis consisted of the mean length of stay, hospital charges, and CMS payments between the two cohorts. The study was deemed exempted by the Institutional Review Board at the University of Texas Medical Branch, Galveston, TX. SAS 9.4 (SAS, Inc., Cary, NC, USA) was used for statistical analyses. A *p*-value < 0.05 was considered the threshold for statistical significance.

## 3. Results

### 3.1. Demographic Characteristics

A total of 13,072 patients identified with DRG 330 were included in this study. The cohorts were divided into the NSC group (*N* = 4324) and the SC group (*N* = 8748) Table 1. The mean age difference of the groups was found to be statistically significant, but unlikely clinically significant (NSC: 76.1 years-old ± 7.5 years vs. SC: 77 years-old ± 7.5 years; <0.0001). There were no statistically significant differences between the groups for gender (*p* = 0.19) or race/ethnicity (*p* = 0.37).

### 3.2. Hospital Characteristics, Discharge Destination, and Inpatient Mortality

The majority of patients were treated at urban hospitals (*N* = 11,479) with no differences between the cohorts in treatment center type (*p* = 0.33) Table 1. However, there were notable differences between the cohorts for discharge destination and inpatient mortality. While the majority of patients were discharged home in each group, the SC group were significantly less likely to be discharged home (NSC—59% vs. SC—48%; *p* = 0.0005). As a result, the SC group required more post-acute care in the form of home health services (NSC—22% vs. SC—24%), SNF placement (NSC—15% vs. SC—21%), or be discharged to a designation classified as ‘other’ (NSC—5% vs. SC—7%; *p* = 0.0005). Among the two cohorts, inpatient mortality was significantly higher in the SC group (1.0% vs. 0.4%; *p* = 0.001).

### 3.3. Colectomies per Provider per Year and Index Hospital Length of Stay (LOS), Charges and Payments

Among providers that perform colectomies, our data demonstrates a statistically significant volume difference between groups (NSC—6.1 ± 4.8 colectomies vs. SC—6.0 ± 4.6 colectomies; 0 < 0.0001). Open colectomies were performed more commonly than laparoscopic resection in both groups (*p* < 0.0001) Table 1. Interestingly, the SC group had both open and laparoscopic approaches coded together more often (13 vs. 3) suggesting a higher conversion rate to open surgery (*p* < 0.0001). The index admission was longer for the SC group (9.1 ± 5.4 days vs. 6.7 ± 3.9 days; *p* < 0.0001). The total charges were also higher for the SC group ($76,378.30 ± $54,348.30 vs. $59,058.00 ± $40,437.70; *p* < 0.0001). Mean actual reimbursements were also higher for the SC group ($12,169.30 ± $8681.10 vs. $11,452.60 ± $7803.10; *p* < 0.0001). Although the cost was proportionally 29% higher for the SC group, the reimbursement was only 6% higher suggesting a gap between cost of resources and provider compensation.

### 3.4. Comorbidities and Complications Post-Admission

All patients had comorbidities at admission that were sufficient to assign them to DRG 330. There were significant differences between the two cohorts with respect to specific pre-existing comorbidities Table 2. Cardiac disease, pulmonary disease, and renal failure were more common in the SC group, whereas solid tumor without metastasis and metastatic cancer were more common in the NSC group. The most common postoperative complication was postoperative ileus, with hemorrhage and postoperative surgical site infections being other significant complications (Figure 1).

## 4. Discussion

The current data confirm, as expected, that the occurrence of post-admission complication significantly increases LOS, charges, and costs. This same impact was demonstrated in our prior work which showed how healthy patients admitted with DRG 331 migrated to the much more costly DRG 330 due to these surgical complications [3]. The current data provides potential guidance for refinement of assignment process to DRG 330 and better align reimbursement. The data in Table 2 describes present on-admission diagnoses associated with the development of postoperative complications. Importantly, common disorders like diabetes, well-managed hypertension, and tumor stage appear to have little impact on complications and cost. The resulting increase in cost for the index hospitalization, the greater use of expensive post-acute care services, and the relative gap in reimbursement suggest that MS-DRG 330 could be refined to better align risk and cost of care in a value-based program. The need for this refinement is supported by the fact that the SC group had 29% higher total charges but only 6% higher reimbursement, suggesting that the additional resource consumption for patient care was not fully compensated.

Price transparency in any economic setting should accurately reflect potential risk associated with patient related factors which drive both the cost of service delivery by the provider and cost to the consumer. The difference between those costs is often inferred as a reflection on quality because a more efficient provider should require fewer resources while still preserving a profit margin. A refinement in payment policy may better align reimbursement with true physiologic risk factors which in turn drive the services rendered and the acquisition costs for resources required for that care [4]. These data are consistent with previous literature confirming that postoperative complications add variable costs to the provider as reflected by higher charges to Medicare and for higher post discharge costs of care [5,6,7]. These data further inform this gap and by refining patient related diagnoses accounting for the gap between resource application and payment (29% vs. 6%) and allow for broader adoption of specific strategies of prehabilitation, pre-operative optimization, and patient centered pre-admission programs dedicated to achieving fewer operative complications and decrease overall hospital costs [8,9,10,11]. Our findings are in concordance with previously published literature regarding the effect of poor preoperative nutrition, anemia, and hematologic dyscrasias on postoperative complications after colectomy [12,13]. Thus, pre-admission optimization offers surgeons the opportunity to address modifiable factors prior to entering the operating room, likely positively impacting healthcare outcomes post-surgery [14,15].

This study is limited by the reliance upon an administrative dataset and the risks associated with inaccurate documentation and coding, as well as missing data (i.e., procedure type code). The data are also limited by the fact that it does not provide true resource acquisition costs borne by the provider, resulting in an inability to calculate profit margin per encounter. Additionally, a large cohort may demonstrate statistical significance when clinically it is irrelevant (i.e., age, colectomies per provider). These limitations highlight potential opportunities for additional economic data that could potentially lead to a provision of a more accurate price transparency tool for beneficiaries. It would also assure that providers experience a resource-based reimbursement model to more effectively compete on quality.

## 5. Conclusions

The use of MS-DRG coding has the potential to provide Medicare fiscal intermediaries, beneficiaries, and providers with a more accurate understanding of the relative impact of their baseline health versus the quality of care provided by an institution. The data strongly suggest that the current system does not adequately align payments and the impact of serious pre-existing co-morbidities on the outcomes for colectomy versus tumor stage and controlled medical illness related factors. Specifically, certain present on admission diagnoses may need to be removed from the “CC” list in the MS-DRG program. The data further suggest that providers may benefit by more fully understanding the cost reduction opportunities resulting from specific preadmission modification of certain illnesses. Therefore, reimbursement may be inadequate to cover the costs of managing both the medical physiology of comorbidities and the typical surgical risks associated with these procedures. Tumor specific characteristics by themselves do not seem to be important cost drivers in the MS-DRG system.

## Figures and Tables

**Figure 1 healthcare-08-00529-f001:**
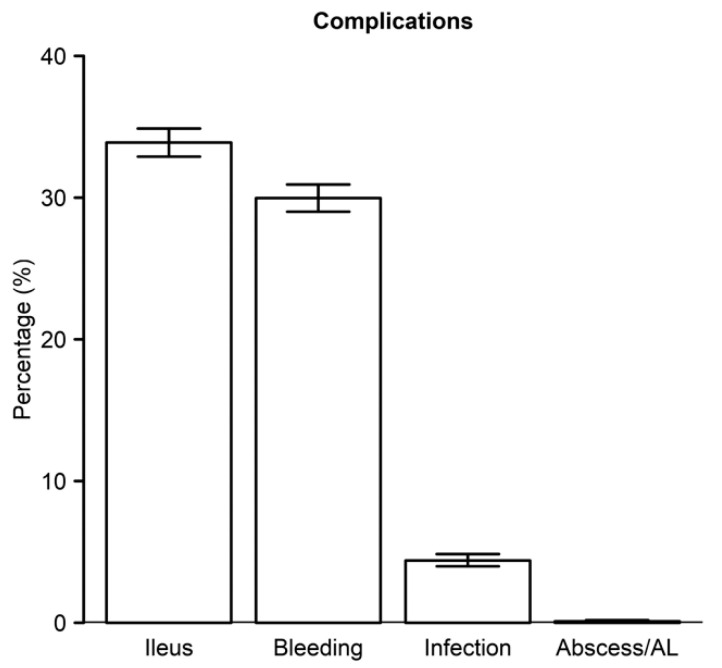
Common post-admission colectomy complications.

**Table 1 healthcare-08-00529-t001:** Patent demographics, procedure type, and index hospital variables.

Characteristics	Absence of Surgical Complications (*N*)	%	Surgical Complications (*N*)	%	*p*-Value
Age	76.1 ± 7.5	-	77.0 ± 7.5	-	<0.0001
Sex					
Men	1557	36	3253	37	0.19
Women	2767	64	5495	63	
Race					
White	3776	87	7603	87	0.37
Black	345	8	762	9	
Hispanic	63	1	108	1	
Other	140	3	275	3	
Procedure Type					
Open	1833	42	4034	46	<0.0001
Laparoscopic	1250	29	2245	26	
Hospital Type					
Rural	544	13	1048	12	0.33
Urban	3780	87	7699	88	
Colectomies Per Provider Per Year	6.1 ± 4.8	--	6.0 ± 4.6	--	<0.0001
Length of Stay	6.7 ± 3.9	--	9.1 ± 5.4		<0.0001
Total Charges ($)	59,058.0 ± 40,437.7	--	76,378.30 ± 54,348.30	--	<0.0001
Payments ($)	11,452.60 ± 7803.10	--	12,169.30 ± 8681.10		<0.0001
Discharge Destination					
Home	2538	59	4206	48	0.0005
Home Health Services	943	22	2107	24	
SNF	636	15	1806	21	
Other	207	5	629	7	
Inpatient Mortality					
No	4305	99.6	8664	99.0	0.001
Yes	19	0.4	84	1.0	

**Table 2 healthcare-08-00529-t002:** Baseline on-admission comorbidity characteristics of cohorts.

Elixhauser Comorbidity Description	Absence of Surgical Complications (*N*)	%	Surgical Complications (*N*)	%	*p*-Value
Congestive Heart Failure	351	8	729	8	0.69
Cardiac Arrhythmia	780	18	1738	20	0.012
Valvular Disease	273	6	669	8	0.006
Pulmonary Circulation Disorders	76	2	206	2	0.029
Peripheral Vascular Disorders	385	9	813	9	0.48
Hypertension Uncomplicated	2576	60	5288	60	0.34
Hypertension Complicated	341	8	977	11	<0.0001
Paralysis	13	0	19	0	0.35
Other Neurological Disorders	110	3	237	3	0.60
Chronic Pulmonary Disease	778	18	1788	20	0.001
Diabetes Uncomplicated	887	21	1811	21	0.82
Diabetes Complicated	101	2	229	3	0.34
Hypothyroidism	730	17	1545	18	0.28
Renal Failure	345	8	977	11	<0.0001
Liver Disease	122	3	269	3	0.45
Peptic Ulcer Disease excluding bleeding	37	1	87	1	0.50
Lymphoma	59	1	83	1	0.039
Metastatic Cancer	1102	25	1414	16	<0.0001
Solid Tumor without Metastasis	2070	48	3803	43	<0.0001
Rheumatoid Arthritis/collagen	151	3	325	4	0.55
Coagulopathy	82	2	170	2	0.89
Obesity	475	11	895	10	0.19
Weight Loss	322	7	737	8	0.06
Fluid and Electrolyte Disorders	638	15	1353	15	0.29
Blood Loss Anemia	174	4	378	4	0.46
Deficiency Anemia	203	5	460	5	0.18
Alcohol Abuse	14	0	40	0	0.31
Drug Abuse	4	0	9	0	1
Psychoses	38	1	59	1	0.23
Depression	384	9	903	10	0.010

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
