# Peer review of "Colectomy among Fee-for-Service Medicare Enrollees Coded as DRG 330: A Potential Platform to Allow Consumer Cost Transparency?"

_healthcare, 2020, doi:10.3390/healthcare8040529_

Round 1

Reviewer 1 Report

The authors have clearly thought about and considered the previous review questions and concerns; and they have made a number of (improving) revisions to the writing as well as the data presentation. Because of their careful responses, I believe that they have addressed all of my concerns enough to warrant acceptance of this manuscript.

Author Response

The authors appreciate the guidance provided by the reviewer’s comments.

Reviewer 2 Report

Overall, an interesting way to look at the impact of complications on reimbursement and quality. The authors responded to the comments appropriately and made significant changes to improve the manuscript. 

My only additional suggestion is for the conclusion- "tumor specific" is repeatedly used, however, not all patients had a diagnosis of colon cancer as the indication for colectomy. I would recommend "disease specific" terminology to avoid confusion. 

Author Response

The authors appreciate the suggestion by the reviewer and have refined the terminology to tumor stage.

Reviewer 3 Report

The revised version describes more clearly the motivation of study. The data provides distinction between the

impact of specific medical complications versus tumor related diagnoses on the development of

postoperative complications and the associated adverse economic impact. But the manuscript still does not adequately explain the significance and innovativeness of the work. What is the new contribution of the study except to highlight the difference between the two groups in one single DRG? The descriptive results of higher inpatient costs and mortality are expected  based on the definition of SC and NSC groups.

The challenging part could be how these were coded but the study does not elaborate on that part. No hypotheses or research questions were provided as these were probably too obvious.

An important finding that can be expanded upon in future could be the nature of patients who are likely to develop higher complications after admission. Studying the characteristics of patients with higher risks versus those with lower risks could have important policy implications in the DRG assignment system of the index admission.

The study highlights the issue of DRG migration and its associated impact on payers versus providers. However,  limiting the work on  only one DRG does not sufficiently guarantee the generalizability and its applicability in a broader context. The study claims that their data shows that MS-DRG 330 conflates the impact of patient physiology

with tumor associated clinical status. Given the validity of this claim, there is no recommendation on how to devise an alternative payment system.

Author Response

The revised version describes more clearly the motivation of study. The data provides distinction between the impact of specific medical complications versus tumor related diagnoses on the development of postoperative complications and the associated adverse economic impact. But the manuscript still does not adequately explain the significance and innovativeness of the work. What is the new contribution of the study except to highlight the difference between the two groups in one single DRG? The descriptive results of higher inpatient costs and mortality are expected based on the definition of SC and NSC groups.

The authors have provided a bit more clarification regarding the existing gap in DRG 330 related to diagnoses more closely aligned with risk and cost.  We have clarified the opportunity for classification by certain more impactful physiologic derangements vs well controlled disorders and tumor stage specific diagnoses. We hope these changes address the reviewer’s concerns.

The challenging part could be how these were coded but the study does not elaborate on that part. No hypotheses or research questions were provided as these were probably too obvious.

The coding criteria are well documented and codified for coders and beyond the capabilities of the data to refine.

An important finding that can be expanded upon in future could be the nature of patients who are likely to develop higher complications after admission. Studying the characteristics of patients with higher risks versus those with lower risks could have important policy implications in the DRG assignment system of the index admission.

We agree with the reviewer as the existing data suggest any postoperative complication post-surgery is directly associated with cost of post-acute care and readmission costs.

 The study highlights the issue of DRG migration and its associated impact on payers versus providers. However,  limiting the work on  only one DRG does not sufficiently guarantee the generalizability and its applicability in a broader context. The study claims that their data shows that MS-DRG 330 conflates the impact of patient physiology with tumor associated clinical status. Given the validity of this claim, there is no recommendation on how to devise an alternative payment system.

The generalizability is a potential limitation of the study; however, it is likely these findings would apply more broadly as the underlying methodology defines certain diagnoses as CC’s regardless of the procedure in question.  We believe the data clarifies a refinement of diagnoses considered to be CC’s may need refinement and this is reflected in our conclusion.

Round 2

Reviewer 3 Report

No further comments.

This manuscript is a resubmission of an earlier submission. The following is a list of the peer review reports and author responses from that submission.

Round 1

Reviewer 1 Report

The manuscript is unclear about the study purpose and motivation. First of all, why is DRG 330 used as a test case? Is it a representative DRG? A more generalized approach involving a number of DRGs may be helpful in deriving any major conclusions.

The authors need to clearly present what they are trying to show. If they are trying to show that cases with post operative complications cause higher cost increases and inpatient mortality than those with baseline complications, isn’t that somewhat expected? What does this actually mean? What is the motivation of the study?

In that context, it may be a good idea to start with some study hypotheses about the difference between the two groups based on previous studies and common knowledge. The authors may then discuss later in the Discussion section whether the findings conform to the expected direction and magnitude of the effects. And if not, why not.

The methods section needs to clearly specify that it is a descriptive comparison between two groups of patients. So the subheadings ‘outcome’ and ‘covariates’ are not appropriate for such comparisons since this is an unadjusted analysis. In this context, the variables chosen for the comparative analysis should be justified.

The conclusion needs to follow from the
findings and the policy implications of the findings should be articulated more clearly. It is not clear what message the paper is providing. Please explain in simple language what your findings mean in terms of CMS DRG classification system. Does DRG system provide adequate cost transparency to be used as a standard classification system?

Author Response

Responses to Reviewer #1

  • The manuscript is unclear about the study purpose and motivation. First of all, why is DRG 330 used as a test case? Is it a representative DRG? A more generalized approach involving a number of DRGs may be helpful in deriving any major conclusions.

The authors thank the reviewer for their comments. This was approach was taken based on prior research studies focused solely on colon surgery outcomes. The authors have experience with using the Medicare Database, as well as DRG 331 and 330. Additionally, colon surgery is a common surgery among colorectal and general surgeons in the U.S. While using additional DRGs could provide a more generalized approach, the authors were focused on areas of their own clinical expertise.

  • The authors need to clearly present what they are trying to show. If they are trying to show that cases with post operative complications cause higher cost increases and inpatient mortality than those with baseline complications, isn’t that somewhat expected? What does this actually mean? What is the motivation of the study?

    The reviewer makes an insightful comment. The authors have modified the introduction beginning on line 47 with a clearer explanation and our motivation for the study:

Individual patients are designated a class on hospital admission, which may be amended prior to discharge based on the post-admission course. The broad assumption of this system is that resource consumption is similar for all patients in a given category.  However, for DRG 330 this assumption risks conflating the resource consumption related to specific present on admission comorbidities with those required to manage common surgical complications.  This distinction regarding the relationship between various comorbidities and complications in this population has not been well studied.

Our group has previously explored this concept by defining the concept of “DRG migration” which is defined by patient assignment to a higher cost DRG due only to post admission comorbidity or complications (CC) [3]. This analysis confirmed the significant cost impact of converting a patient from DRG 331 to the more highly reimbursed DRG 330 due to surgical complications from the perspective of the payor, however the provider was better reimbursed to manage those surgical complications.  This results in a potential value gap where providers with lower complications are reimbursed less whereas providers with higher complication rates may benefit economically.

The current study expands on this concept by attempting to define payment variations among patients with significant comorbidities present on admission to warrant DRG 330 assignment and define the relative impact of surgical complications on the cost of care. Therefore, we assessed clinical outcomes and cost of care for this defined population based on: 1) the absence of surgical complications (NSC); or 2) the development of surgical complications (SC).

3) In that context, it may be a good idea to start with some study hypotheses about the difference between the two groups based on previous studies and common knowledge. The authors may then discuss later in the Discussion section whether the findings conform to the expected direction and magnitude of the effects. And if not, why not.

The reviewer makes an important point in demonstrating common knowledge and understanding of the groups based on previous studies. This was also included in response to question #2. Additionally, starting at line 150 in the Discussion Section, the authors have added the following:
“The current data confirm
, as expected, that the occurrence of post-admission complication significantly increases LOS, charges and costs. This same impact was demonstrated in our prior work which showed how healthy patients admitted with DRG 331 migrated to the much more costly DRG 330 due to these surgical complications [3]. The current data provides a distinction between the impact of specific medical complications versus tumor related diagnoses on the development of postoperative complications and the associated adverse economic impact. The resulting increase in cost for the index hospitalization, the greater use of expensive post-acute care services, and the relative gap in reimbursement suggest that MS-DRG 330 conflates the impact of patient physiology with tumor associated clinical status. Our study reveals the latter may not be associated with resource consumption in this DRG during the index admission. Interestingly, the SC group had 29% higher total charges but only 6% higher reimbursement, suggesting that the additional resource consumption for patient care was not fully compensated.”

4) The methods section needs to clearly specify that it is a descriptive comparison between two groups of patients. So the subheadings ‘outcome’ and ‘covariates’ are not appropriate for such comparisons since this is an unadjusted analysis. In this context, the variables chosen for the comparative analysis should be justified.

The authors have modified the methods section:

1) Under subsection 2.1, the following sentence has been modified to specify the descriptive nature of the group comparison on line 60:
“A descriptive comparison between two groups, designed as a retrospective cohort study using the 5% random sample Medicare Database from 2011 to 2014 was completed.”

2) The subheadings ‘outcome’ and ‘covariates’ have been removed.

5)The conclusion needs to follow from the findings and the policy implications of the findings should be articulated more clearly. It is not clear what message the paper is providing. Please explain in simple language what your findings mean in terms of CMS DRG classification system. Does DRG system provide adequate cost transparency to be used as a standard classification system?
The authors have revised the manuscript and have modified the entire Conclusion Section in order to articulate a clear message to the readers.

Conclusion Section
The use of CMS-DRG coding has the potential to provide Medicare fiscal intermediaries, beneficiaries, and providers with a more accurate understanding of the relative impact of their baseline health versus the quality of care provided by an institution.  The data strongly suggest that the current system does not adequately account for the impact of serious pre-existing co-morbidities on the outcomes for colectomy versus tumor-related factors.  The data further suggest that providers may benefit by more fully understanding the cost of preventive measures as a means of reducing total cost of care for this population. Therefore, reimbursement may be inadequate to cover the costs of managing both the medical physiology of comorbidities and the typical surgical risks associated with these procedures.  Tumor specific characteristics by themselves do not seem to be important cost drivers in the MS-DRG system.

Reviewer 2 Report

The article presents an interesting case on the need to rethink the way in which patients are classified and providers are paid using DRG in th US.

Minor comments are attached in the manuscript. The main issue is to better clarify the problem, the aim of the article, and the recommendations/ conclusions coming from the results.

Author Response

Responses to Reviewer #2
The article presents an interesting case on the need to rethink the way in which patients are classified and providers are paid using DRG in th US.

Minor comments are attached in the manuscript. The main issue is to better clarify the problem, the aim of the article, and the recommendations/ conclusions coming from the results.

  1. The abstract has been reformatted. It now reads: Length of stay (LOS)
  2. The differences between DRG 330 and DRG 329 are pre-defined by the Centers for Medicare and Medicaid Services. In general, those classified as DRG 329 are sicker patients.
  3. Your understanding of the study to this point is well explained. To better aid our readers, we have added the following in the Methods Section
    “For this study, we included all patients assigned to DRG 330 due to their baseline comorbidities. Two groups of patients were then assessed based on: 1) the absence of surgical complications (NSC); or 2) the development of surgical complications (SC). All patients classified as DRG 331 or DRG 329 upon discharge were excluded.”
  4. Additionally, the authors have modified the Introduction section starting on line 54 to make this point clearer:
    “Additionally, the authors modified the Introduction Section starting on line 54:
    Our group has previously explored this concept by defining the concept of “DRG migration” which is defined by patient assignment to a higher cost DRG due only to post admission comorbidity or complications (CC) [3].
    This analysis confirmed the significant cost impact of converting a patient from DRG 331 to the more highly reimbursed DRG 330 due to surgical complications from the perspective of the payor, however the provider was better reimbursed to manage those surgical complications.  This results in a potential value gap where providers with lower complications are reimbursed less whereas providers with higher complication rates may benefit economically.” 
  5. ‘Other’ is pre-defined by the Centers for Medicare and Medicaid Services within the MEDPAR files.
  6. Due to relatively small numbers by year, this data was presented in the aggregated format to keep consistent with other studies in the surgical literature. We did not look at geographical location in this study, but the reviewer makes an interesting point and it is a factor we could look at in future studies.
  7. No, the authors did not control for other hospital variables. However, understanding this for future studies makes for an interesting study question.
  8. Only post-admission complications that occurred during the index admission were examined.
  9. Table 3 has been removed.
  10. The authors have removed this part of the Discussion Section. Instead, we discuss potential policy implications elsewhere in the Discussion Section as follows:
    “The present study suggests that patients with comorbidities that impact their physiology may warrant a different payment classification than patients with tumor characteristics that do not alter the patient’s response to surgery. A refinement in payment policy may better align reimbursement with services rendered and the acquisition costs for resources required for that care.
    [4].”
  11. The authors have revised several sections to further explain the connection between the potential use of the DRG classification system and price transparency, hopefully to the satisfaction of the reviewer.

Reviewer 3 Report

In this manuscript, the authors utilize the CMS-DRG codes, specifically DRG 330, for colectomies to infer quality. Overall, I found this to be a very interesting and creative way to look at cost of care and its effect of post operative complications for colectomies. It is not surprising that cost increases with post operative complications, however, it is interesting to see that pre-existing or co-morbid conditions did not significantly affect the risk of a complication in patients undergoing a colectomy.

1) Can the authors infer from the data gathered what are the risk factors driving this difference in quality? It doesn't seem to be related to surgeon volume or the type of hospital. Can further details be provided of the diagnosis and how this differed in each group? Do the authors believe that some diagnoses,i.e the reason for colectomy (cancer- colon vs rectal, IBD, diverticulitis) plays a role?

2) Does your dataset allow you to link the index admission with readmissions to the hospital within 30 days? If so, was there a difference?

Author Response

Responses to Reviewer #3
In this manuscript, the authors utilize the CMS-DRG codes, specifically DRG 330, for colectomies to infer quality. Overall, I found this to be a very interesting and creative way to look at cost of care and its effect of post operative complications for colectomies. It is not surprising that cost increases with post operative complications, however, it is interesting to see that pre-existing or co-morbid conditions did not significantly affect the risk of a complication in patients undergoing a colectomy.

1) Can the authors infer from the data gathered what are the risk factors driving this difference in quality? It doesn't seem to be related to surgeon volume or the type of hospital. Can further details be provided of the diagnosis and how this differed in each group? Do the authors believe that some diagnoses,i.e the reason for colectomy (cancer- colon vs rectal, IBD, diverticulitis) plays a role?

The authors have modified the Discussion Section beginning on line 165

“The present study suggests that patients with comorbidities that impact their physiology may warrant a different payment classification than patients with tumor characteristics that do not alter the patient’s response to surgery. A refinement in payment policy may better align reimbursement with services rendered and the acquisition costs for resources required for that care. [4]. These data are consistent with previous literature confirming that postoperative complications add variable costs to the provider as reflected by higher charges to Medicare and for higher post discharge costs of care [5-7].  However, the gap between charges as a surrogate for resource application and payment (29% vs. 6%) suggests that MS-DRG 330 may need refinement of the diagnoses used for assignment to the group to better reflect the inherent different in the risk of complications.

The data further suggest that in patients with specific chronically comorbid conditions, there may be a need to compensate for specific strategies of prehabilitation, pre-operative optimization, and patient centered pre-admission programs dedicated to achieving fewer operative complications and decrease overall hospital costs [8-11]. Our findings are in concordance with previously published literature regarding the effect of poor preoperative nutrition, anemia, and hematologic dyscrasias on postoperative complications after colectomy [12, 13]. Thus, pre-admission optimization offers surgeons the opportunity to address modifiable factors prior to entering the operating room, likely positively impacting healthcare outcomes post-surgery [14, 15]. However, it appears that the tumor stage itself has little impact on the risk of an elective curative colon resection. Therefore, tumor specific characteristics by themselves do not seem to be important cost drivers in the MS-DRG system.”

The authors have previously examined the outcomes of patients with benign vs. cancer undergoing elective colon resection using DRG 331 and DRG 330. In brief, tumor-related diagnoses (e.g. oncological group) resulted in less charges and LOS than those with benign diseases (i.e. IBD, diverticulitis). While we did not specifically examine benign diseases vs cancer in this study, we believe the findings would have been similar if a sub-analysis was performed in this cohort.

Hughes BD, Hancock KJ, Shan Y, Thakker RA, Maharsi S, Tyler DS, Mehta HB, Senagore AJ. Cost of benign versus oncologic colon resection among fee-for-service Medicare enrollees. J Surg Oncol. 2019 Aug;120(2):280-286. doi: 10.1002/jso.25511. Epub 2019 May 27. PMID: 31134661; PMCID: PMC6635007.

2) Does your dataset allow you to link the index admission with readmissions to the hospital within 30 days? If so, was there a difference?
The reviewer asks excellent questions here regarding the perioperative period of patients. However, the authors only looked at index admissions.

Reviewer 4 Report

In writing this review of the manuscript by Hughes et al submitted to Healthcare, I re-read the manuscript multiple times on several different occasions. I have a number of thoughts and questions both general and specific.

1) Lack of clarity in writing

The manuscript is extremely difficult to understand, with confusing writing, inconsistencies, and apparently contradictory statements. Despite reading through the manuscript multiple times, it is still unclear to me what the authors are actually trying to demonstrate or explore, or what question they are trying to answer with these data and this set of analyses. Although I could eventually understand the comparisons that were being examined (for the most part, but see below), I could not tie the data and analyses presented to the topic of price transparency. Although as a health economist I have examined resource use, cost, cost assessment and the like in a number of different settings, I was not able to grasp what the authors were trying to show. I cannot tell if it is because the writing was confusing to me, or because there really was a lack of connection between the analyses and the underlying research question.

For example, sentences such as this one created confusion:

“At this moment in time, there is no consumer level data that transparently provides cost data for a beneficiary to understand both the impact of neither their underlying health nor the potential cost/quality benefits across providers.”

But as one result of this confusion, I am unable to judge whether or not the manuscript asks (or answers) an important question, which unfortunately is a stumbling block for my providing constructive criticism or advice on the overall merits of the research. However, I hope that the more specific concerns addressed below will be helpful to the authors.

2) Cohort construction

The cohort is supposed to consist of all (and only) patients with a DRG 330, which indicate patients with a comorbidity or complication (but not a major comorbidity of complication). However, because the writing is confusing, there are a number of things are that unclear about this cohort.

  1. a) Is the cohort restricted just to those patients who had at least one comorbidity? Or can a patient get into the cohort if they have a complication during the inpatient stay? (I believe that it is the former, but it is hard to tell.)
  2. b) If the cohort is restricted just to those with at least one comorbidity, does that comorbidity have to be present at admission? It was not until the first sentence in section 3.4 that it because clear that the answer to this question is YES.

3) Small numbers

In the narrative related to Table 1 and in both Tables 2 and 3, there is inclusion of numbers and statistical tests when the sample sizes are too small to be meaningful or interesting. For example, the 3 vs 13 patients mentioned in the table 1 narrative. In addition, for several of the rows in table 2, and most of the rows in table 3, the numbers are too small to be able to say anything about them – even if a statistical test CAN be performed and seems significant; the numbers and statistical tests are not meaningful.

4) Small differences

In a number of the results/analyses, a significant difference is found even though the absolute difference between comparison groups is very small. The authors rightly point out that the significant difference found between the provider volume of colectomies performed is not clinically relevant, the same is true for other instances where the difference is quite small, but statistically significant only because the sample size is large. For example, the mean ages of the groups were 76.1 (sd 7.5) and 77.0 (sd 7.5).

5) Table 3

Even after multiple attempts, I cannot figure out what information is trying to be presented in Table 3, much less which statistical test was done.

6) Unadjusted analyses

The analyses do not adjust for any patient demographics or illness burden; in comparisons where we would predict that these cofactors would have an effect. Therefore, it is unclear that conclusions can be drawn from the analyses presented.

7) Table 1 questions

  1. a) The percentages for “Procedure type” in table 1 do not sum to 100; and it is unclear what the numbers and percentages mean or how they are defined.
  2. b) For “Discharge destination”, the narrative is confusing in saying that 59% of no post admission complications (PAC) patients and 48% of PAC patients were discharged “Home”, when the next sentence indicates that patients in the Home Health Services category were also discharged home.

Author Response

Responses to Reviewer #4
In writing this review of the manuscript by Hughes et al submitted to Healthcare, I re-read the manuscript multiple times on several different occasions. I have a number of thoughts and questions both general and specific.

1) Lack of clarity in writing

The manuscript is extremely difficult to understand, with confusing writing, inconsistencies, and apparently contradictory statements. Despite reading through the manuscript multiple times, it is still unclear to me what the authors are actually trying to demonstrate or explore, or what question they are trying to answer with these data and this set of analyses. Although I could eventually understand the comparisons that were being examined (for the most part, but see below), I could not tie the data and analyses presented to the topic of price transparency. Although as a health economist I have examined resource use, cost, cost assessment and the like in a number of different settings, I was not able to grasp what the authors were trying to show. I cannot tell if it is because the writing was confusing to me, or because there really was a lack of connection between the analyses and the underlying research question.

The authors thank the reviewer for the detailed assessment of the manuscript.

For example, sentences such as this one created confusion:

“At this moment in time, there is no consumer level data that transparently provides cost data for a beneficiary to understand both the impact of neither their underlying health nor the potential cost/quality benefits across providers.”

The paragraph related to consumer level data has been modified:
Price transparency in any economic setting should accurately reflect both the cost of service delivery by the provider and cost to the consumer receiving those benefits. The difference between those costs is a reflection on quality because a more efficient provider should require fewer resources while still preserving a profit margin. The present study suggests that patients with comorbidities that impact their physiology may warrant a different payment classification than patients with tumor characteristics that do not alter the patient’s response to surgery. A refinement in payment policy may better align reimbursement with services rendered and the acquisition costs for resources required for that care. [4]. These data are consistent with previous literature confirming that postoperative complications add variable costs to the provider as reflected by higher charges to Medicare and for higher post discharge costs of care [5-7].  However, the gap between charges as a surrogate for resource application and payment (29% vs. 6%) suggests that MS-DRG 330 may need refinement of the diagnoses used for assignment to the group to better reflect the inherent different in the risk of complications.”

But as one result of this confusion, I am unable to judge whether or not the manuscript asks (or answers) an important question, which unfortunately is a stumbling block for my providing constructive criticism or advice on the overall merits of the research. However, I hope that the more specific concerns addressed below will be helpful to the authors.

The authors have made numerous revisions throughout the manuscript to enhance the quality of the manuscript.

2) Cohort construction

The cohort is supposed to consist of all (and only) patients with a DRG 330, which indicate patients with a comorbidity or complication (but not a major comorbidity of complication). However, because the writing is confusing, there are a number of things are that unclear about this cohort.

  1. a) Is the cohort restricted just to those patients who had at least one comorbidity? Or can a patient get into the cohort if they have a complication during the inpatient stay? (I believe that it is the former, but it is hard to tell.)

The cohort consists of all patients classified as DRG 330 due to baseline comorbidities. Additionally, the authors have modified the group names (below) to make it more clear to the readers:

 The authors have modified the following within the manuscript to make it clear for all readers:

Introduction Section (starting at line 62)

“The current study expands on this concept by attempting to define payment variations among patients with significant comorbidities present on admission to warrant DRG 330 assignment and define the relative impact of surgical complications on the cost of care. Therefore, we assessed clinical outcomes and cost of care for this defined population based on: 1) the absence of surgical complications (NSC); or 2) the development of surgical complications (SC).

Methods Section: (starting at line 78)
“For this study, we included all patients assigned to DRG 330 due to their baseline comorbidities. Two groups of patients were then assessed based on: 1) the absence of surgical complications (NSC); or 2) the development of surgical complications (SC). All patients classified as DRG 331 or DRG 329 upon discharge were excluded.”

  1. b) If the cohort is restricted just to those with at least one comorbidity, does that comorbidity have to be present at admission? It was not until the first sentence in section 3.4 that it because clear that the answer to this question is YES.

Yes. The authors have modified (see changes under question 1a) the manuscript to make this clear for all readers much earlier in the manuscript.

3) Small numbers

In the narrative related to Table 1 and in both Tables 2 and 3, there is inclusion of numbers and statistical tests when the sample sizes are too small to be meaningful or interesting. For example, the 3 vs 13 patients mentioned in the table 1 narrative. In addition, for several of the rows in table 2, and most of the rows in table 3, the numbers are too small to be able to say anything about them – even if a statistical test CAN be performed and seems significant; the numbers and statistical tests are not meaningful.
The authors agree that some of the statistical significance findings do not have clinical relevance. While 3 and 13 patients may be small, the data does support the notion that in those with postoperative complications, more converted from a laparoscopic surgery to an open surgery suggestive of a difficult operation; therefore, more likely to be associated with a complication.

Table 3 has been removed.

 4) Small differences

In a number of the results/analyses, a significant difference is found even though the absolute difference between comparison groups is very small. The authors rightly point out that the significant difference found between the provider volume of colectomies performed is not clinically relevant, the same is true for other instances where the difference is quite small, but statistically significant only because the sample size is large. For example, the mean ages of the groups were 76.1 (sd 7.5) and 77.0 (sd 7.5).
The authors modified the limitation section on lines 212-214 to include: “Additionally, a large cohort may demonstrate statistical significance when clinically it is irrelevant (i.e. age, Colectomies Per Provider).

5) Table 3

Even after multiple attempts, I cannot figure out what information is trying to be presented in Table 3, much less which statistical test was done.
The authors have removed Table 3. 

6) Unadjusted analyses

The analyses do not adjust for any patient demographics or illness burden; in comparisons where we would predict that these cofactors would have an effect. Therefore, it is unclear that conclusions can be drawn from the analyses presented.

The authors understand the reviewer’s concerns. However, in terms of design we looked at baseline comorbidities compared to those who developed post-admission complications on a variable-by-variable basis. We believe this could have been problematic if the baseline characteristics differed in a meaningful way (i.e. age, sex, race, for instance). If any of these variables had differed, we believe it would have been a good idea to model the incidence of each complication while adjusting for those baseline variables. Again, while age and colectomies-per-provider-per-year were statistically significant, these were not practical differences to influence clinical outcomes. In other words, we believe the differences to be a product of a large N, and multivariable modeling was not performed.  

Additionally, the authors have modified the Conclusion Section to make reasonable conclusive statements based on the information we do have.

7) Table 1 questions

  1. a) The percentages for “Procedure type” in table 1 do not sum to 100; and it is unclear what the numbers and percentages mean or how they are defined.

Due to missing data, the sum does not equal 100; Approximately 30 % (29 % and 28 %, respective to the cohorts as presented in the table) had missing procedure code data. Therefore, this data was omitted. Despite this, the data presented aligns with previous surgical literature in that more open colectomies have been performed compared to laparoscopic colectomies in both cohorts.

The authors have added the following in the limitation section:
This study is limited by the reliance upon an administrative dataset and the risks associated with inaccurate documentation and coding, as well as missing data (i.e. procedure type code).

  1. b) For “Discharge destination”, the narrative is confusing in saying that 59% of no post admission complications (PAC) patients and 48% of PAC patients were discharged “Home”, when the next sentence indicates that patients in the Home Health Services category were also discharged home.

In the context of Medicare, ‘home’ means the patient was discharged from the hospital with the expectation of being able to perform their activities of daily living independently. In contrast, most times, those who required home services did not meet the threshold for additional services offered by  SNF or some other rehabilitation facility.  

The authors have modified the sentences beginning on line 117:
While the majority of patients were discharged home in each group, the SC group were significantly less likely to be discharged home (NSC-59% vs. SC-48%; p=0.0005).  As a result, the SC group required more post-acute care in the form of home health services (NSC-22% vs. SC-24%), SNF placement (NSC-15% vs. SC- 21%), or be discharged to a designation classified as ‘other’ (NSC- 5% vs. SC7%; p=0.0005). …”